# Deep Heterogeneity: A New Paradigm for Time Series Forecasting

## Abstract

Time series forecasting frequently employs signal decomposition to disentangle complex series into simpler, more structured components. However, a prevalent limitation within this paradigm is the principle of Shallow Heterogeneity, where structurally homogeneous processing blocks are indiscriminately applied to signal components with fundamentally different properties. This misalignment between a model's fixed inductive biases and the heterogeneous nature of the decomposed signals imposes a significant bottleneck on forecasting accuracy. To address this challenge, we propose a novel framework centered on the principle of Deep Heterogeneity, instantiated in our model, HeteroMixer. Our model first decomposes the series using a wavelet transform and then introduces a deeply heterogeneous architecture, featuring a powerful, dual-domain Trend-Seasonality Extractor (TSE) specifically for the vital low-frequency trend, alongside lightweight expert modules for high-frequency details. Furthermore, we replace static signal reconstruction with a novel Hybrid Prediction Framework, where an attention-based Multi-scale Fusion Transformer (MFT) learns to adaptively predict a corrective residual to a stable, classical baseline. Extensive experiments on seven challenging benchmarks demonstrate that HeteroMixer establishes a new state-of-the-art, significantly outperforming existing methods. Our work validates the thesis that deeply aligning specialized, heterogeneous architectural biases with the intrinsic properties of signal components is a crucial strategy for advancing forecasting accuracy.

## 1 Introduction

Long-term Time Series Forecasting (LTSF) is a task of paramount importance, serving as the predictive engine for decision-making in a multitude of critical, high-impact domains.He (2023); Kong et al. (2025) From anticipating electricity consumption for national grid stability and forecasting financial market volatility for risk management, to modeling climate patterns and predicting traffic flow for urban planning, the ability to produce accurate and reliable long-range forecasts is not merely an academic pursuit but a cornerstone of modern operational intelligence. A dominant and empirically successful paradigm in modern LTSF is signal decomposition, where state-of-the-art models such as FEDformerZhou et al. (2022) and DLinearZeng et al. (2023) explicitly disentangle a time series into constituent components. The core assumption is that modeling these simpler, more structured parts individually is a more tractable problem than modeling the complex raw series in its entirety. This approach has undeniably advanced the frontier of forecasting accuracy.

However, a critical analysis of this paradigm reveals a persistent, yet subtle, limitation that curtails its full potential. While these models conceptually separate the signal, they often fail to follow through with true architectural specialization, a practice we term Shallow Heterogeneity. Existing approaches typically employ structurally homogeneous processing blocks for both low- and high-frequency components, creating a fundamental misalignment between the fixed inductive biases of the architecture and the intrinsically heterogeneous properties of the decomposed signals.

This heterogeneity is not merely theoretical; it is vividly revealed by a standard wavelet decomposition of a real-world time series, as shown in Figure 1. The low-frequency approximation coefficients (cA) exhibit a clear, smooth, and slowly-varying structure, embodying the core trend. In stark contrast, the high-frequency detail coefficients (cD) are characterized by high volatility and transient,

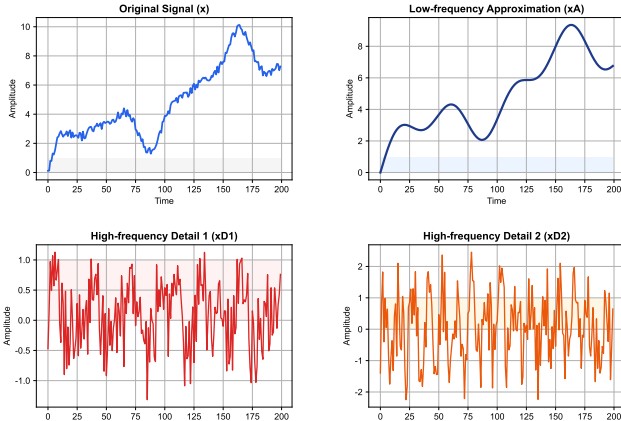

Figure 1: Wavelet Decomposition of Time Series Revealing the heterogeneous nature of decomposed signal components

spike-like patterns. Applying a single, homogeneous architecture to these starkly different components is inherently suboptimal: a complex model risks overfitting the smooth trend, while a simpler one fails to capture the rich, non-linear dynamics of the details. This is the central challenge we aim to solve. To overcome the limitations of Shallow Heterogeneity and realize a true Deep Heterogeneity framework, we designed HeteroMixer , an architecture built upon three synergistic principles of innovation.

First, to ensure architectural alignment with signal properties, we practice heterogeneous specialization. After an initial signal disentanglement using the Discrete Wavelet Transform (DWT), we route the resulting components to distinct, purpose-built modules. The low-frequency component is processed by a Trend-Seasonality Extractor (TSE), which integrates Fourier analysis into a Mixer architecture to robustly model global trends. Concurrently, each high-frequency component is handled by a dedicated Stochastic Component Modeler (SCM), a lightweight Mixer optimized for capturing local, volatile patterns.

Second, we move beyond static, rule-based signal reconstruction by introducing adaptive fusion. We argue that the fixed mathematical formula of the inverse wavelet transform is a rigid bottleneck that combines predictive information agnostically. We replace this with our novel Multi-scale Fusion Transformer (MFT), a dynamic, attention-based module that learns to optimally weigh and combine the outputs from our heterogeneous branches, treating them as a sequence of expert opinions to be intelligently synthesized.

Finally, we embed these innovations within a novel Hybrid Prediction Framework to maximize both performance and stability. Instead of completely abandoning the classical inverse transform, we strategically repurpose it to create a robust Stability Baseline ($y_{baseline}$). Our powerful MFT is then tasked with the more refined objective of learning the Corrective Residual ($y_{correction}$) needed to elevate the baseline forecast to the ground truth. The final prediction is the synthesis of these two streams: $y = y_{baseline} + y_{correction}$. This strategy synergizes the reliability of a classical foundation with the deep, corrective power of a data-driven fusion model.

Our contributions are summarized as follows:

- We identify and formalize "Shallow Heterogeneity" as a key limitation in the current state-of-the-art signal decomposition paradigm for LTSF.

- We propose HeteroMixer , a novel architecture that instantiates a "Deep Heterogeneity" framework through principled decomposition and heterogeneous specialization with tailored modules (TSE and SCMs).

- We introduce the Multi-scale Fusion Transformer (MFT), a dedicated attention-based module for adaptive fusion that replaces static reconstruction with a dynamic, learnable synthesis process.

- We design a novel Hybrid Prediction Framework that synergizes a stable baseline forecast with a corrective residual learned by the MFT, enhancing both accuracy and training stability.Extensive experiments on seven challenging benchmarks, including Traffic, Electricity, Weather, and the ETT dataset family, confirm that HeteroMixer establishes a new state-of-the-art, validating the superiority of our proposed framework.

## 2 RELATED WORK

Our work is grounded in several key directions that have recently witnessed remarkable advancements in the field of Long-Term Time Series Forecasting (LTSF), and on this basis, we have conducted in-depth critical analysis and innovation. This section will review these related works and elaborate on how our method overcomes their inherent limitations.

**Deep Learning and Long-Term Time Series Forecasting** In recent years, deep learning models, particularly architectures represented by the Transformer, have achieved a dominant position in LTSF tasks. ModelsLiu et al. (2024); Zhang & Yan (2022), such as InformerZhou et al. (2021), AutoformerWu et al. (2021), and FEDformer address the challenges of long sequences through various mechanisms. However, despite the success of TransformerVaswani et al. (2017) models, their core self-attention mechanism still has a fundamental contradiction with the essence of time series data. The self-attention mechanism is "permutation-invariant"—it treats the input sequence as an unordered set, which makes it difficult to capture the intrinsic relationships in time series data that strictly depend on temporal order. A study tested the performance of Transformer models by shuffling time series data and found that their performance was barely affected, whereas a simple linear model suffered a significant decline. This phenomenon profoundly indicates that correct inductive bias is crucial for time series forecasting. Models like DLinear have proven this point: although their architecture is simple, they can outperform complex Transformer models due to their explicit modeling of time series decomposition. This provides a direct insight for our work: future research should not blindly pursue model complexity, but rather focus on designing inductive biases that are more aligned with the characteristics of time series data.

**Limitations of Time Series Decomposition** Time series decomposition has become an effective paradigm for solving LTSF problemsMurad et al. (2025); Wang et al. (2022). This method decouples complex time series signals into sub-components that can be modeled separately, such as trend, seasonality, and residual. DLinear has achieved remarkable results by modeling the decomposed components through simple linear layers. FEDformer, on the other hand, uses a module called Mixture of Experts Decomposition (MOEDecomp) to handle the trend component and employs a frequency-enhanced Transformer to model the seasonal component. Nevertheless, this "shallow heterogeneous" paradigm still has fundamental flaws. Although the signals are separated, there is a significant bias in the modeling capability across different components. StudiesHan et al. (2025) have found that under end-to-end training, the loss on the trend component is typically 2 to 5 times higher than that on the seasonal component, and the error in trend prediction can account for approximately 80% of the total prediction error. This finding strongly suggests that the modeling of the trend component is the "Achilles' heel" of existing decomposition-based models. Our work aims to fundamentally address this modeling bottleneck through a specially designed "Time-Frequency Mixture of Experts" module, thereby overcoming the inherent limitations of existing decomposition models.

**Time-Frequency Analysis and Adaptive Fusion** Time-frequency analysis offers another promising approach for time series forecasting. The Fourier Transform can effectively capture the global periodicity of signals and has been utilized by models such as FEDformerZhou et al. (2022) to enhance their attention mechanisms. However, for non-stationary signals, the Fourier Transform lacks temporal localization capability. In contrast, the Wavelet Transform has become an ideal tool for processing non-stationary time series data due to its ability to provide both temporal and frequency domain resolution. Studies have shown that incorporating the Wavelet Transform can significantly improve prediction quality. Recently, research has begun to explore the combination of Wavelet Transform with deep learning. For instance, the WaveTS-MZhou et al. (2025) model integrates the Wavelet Transform with a Multilayer Perceptron (MLP) and a Mixture of Experts framework to effectively handle multi-channel dependencies. Building on this, our method further deepens the integration by fusing multiple inductive biases (temporal domain, frequency domain, and deep non-

linearity) into a single expert module tailored for trend modeling, thereby achieving more powerful specialized modeling capabilities. Beyond single-modal modeling, cutting-edge research has also started to explore the adaptive fusion of multi-modal information to enhance model robustness. For example, the T3TimeChowdhury et al. (2025) framework proposes a mechanism for fusing three modalities—temporal domain, frequency domain, and text prompts—and utilizes a "horizon-aware gating mechanism" to learn dynamic adjustment of the priority of temporal and frequency domain features based on the forecasting horizon. Similarly, the AMDHu et al. (2025) model employs an "Adaptive Multi-predictor Synthesis" module to dynamically weight and aggregate multi-scale predictions. These works indicate that learnable reconstruction for decomposed or multi-modal predictions has become an important research direction, which goes beyond the traditional simple linear summation. Our "adaptive fusion mechanism" aligns with this trend, aiming to provide a more advanced, context-aware synthesis strategy to avoid the limitations of "static fusion."

# 3 APPROACH

Our proposed model, HeteroMixer, is designed to instantiate the principle of Deep Heterogeneity. It systematically addresses the challenges of long-term time series forecasting through a progressive, three-stage pipeline: signal disentanglement via wavelet transform, heterogeneous specialization with expert modules, and a novel hybrid prediction framework for intelligent synthesis. The overall architecture is illustrated in Figure 2.

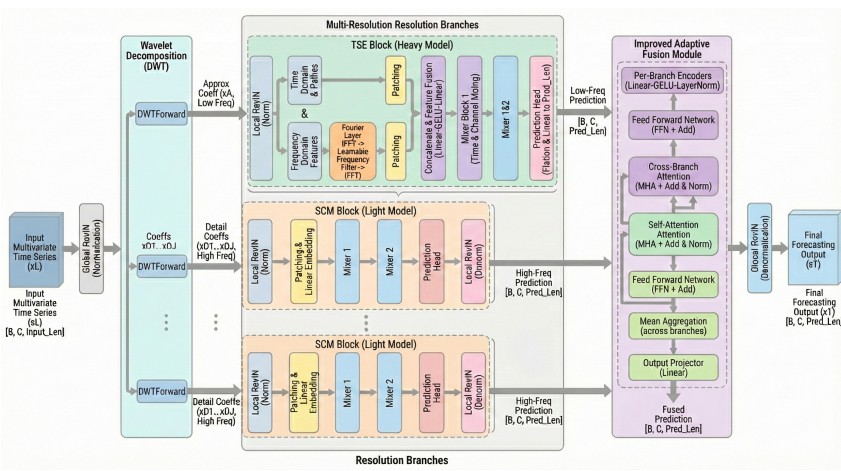

Figure 2: Overall Architecture of HeteroMixer. The input is decomposed via DWT into cA$_J$ (low-frequency trend) and {cD$_j$} (high-frequency details). These components are processed by specialized experts: the dual-domain TSE for cA$_J$ and lightweight SCMs for {cD$_j$}. The Hybrid Prediction Framework synthesizes their outputs via two streams—a baseline from iDWT and a learned correction from MFT—to produce the final forecast $\hat{\mathbf{Y}}$.

## 3.1 PROBLEM FORMULATION AND WAVELET DECOMPOSITION

Let $\mathbf{X} \in \mathbb{R}^{L \times C}$ denote the historical observed time series, where $L$ is the lookback window length and $C$ is the number of variables. Our objective is to predict the future sequence $\mathbf{Y} \in \mathbb{R}^{H \times C}$, where $H$ is the forecasting horizon. The core challenge in long-term forecasting lies in the heterogeneous nature of time series dynamics: global trends evolve slowly and smoothly, while local fluctuations are volatile and stochastic. To address this heterogeneity, HeteroMixer first decomposes the input into hierarchical frequency components, then matches each component with a specialized architecture tailored to its intrinsic properties.

We employ the Discrete Wavelet Transform (DWT) to perform this decomposition. Unlike methods that rely on moving averages or simple filtering, DWT provides a mathematically rigorous multi-resolution analysis that naturally separates signals based on their frequency characteristics while preserving temporal localization—a critical property for non-stationary time series. Given a decom-

position level $J$, DWT recursively applies a pair of complementary filters—a low-pass filter $h$ and a high-pass filter $g$—to decompose the signal. At each level $j \in \{1, \ldots, J\}$, the approximation coefficients from the previous level are split into coarser approximation $cA_j$ and detail coefficients $cD_j$:

$$
\begin{aligned}
cA_j &= \text{DownSample}(cA_{j-1} * h, 2), \\
cD_j &= \text{DownSample}(cA_{j-1} * g, 2),
\end{aligned}
\tag{1}
$$

where $cA_0 = \mathbf{X}$, $*$ denotes convolution, and $\text{DownSample}(\cdot, 2)$ performs downsampling by a factor of 2. This process yields a hierarchical decomposition $\mathcal{C} = \{cA_J, cD_J, cD_{J-1}, \ldots, cD_1\}$, where $cA_J \in \mathbb{R}^{(L/2^J) \times C}$ captures the low-frequency approximation (global trend and seasonality), while $cD_j \in \mathbb{R}^{(L/2^j) \times C}$ for $j = 1, \ldots, J$ represents high-frequency details at different scales.

As visualized in Figure 1, this decomposition reveals stark differences in signal properties. The low-frequency component $cA_J$ exhibits smooth, slowly-varying patterns ideal for trend modeling, while the high-frequency components $\{cD_j\}$ exhibit volatile, spike-like behaviors characteristic of stochastic noise. This observation motivates our central thesis: applying a single, homogeneous architecture to these fundamentally different components—a practice we term Shallow Heterogeneity—is inherently suboptimal. Instead, we must match specialized architectures to specialized signal properties, a principle we call Deep Heterogeneity.

## 3.2 HETEROGENEOUS EXPERT MODULES

Following decomposition, each component is routed to a distinct expert module specifically designed to match its intrinsic statistical properties. This stage embodies the core of our Deep Heterogeneity principle, where architectural specialization replaces structural homogeneity.

The low-frequency component $cA_J$ contains the most critical information about long-term trends and dominant seasonal patterns. Modeling this component accurately is paramount, as prior work has shown that trend prediction errors can account for up to 80% of total forecasting error. To address this challenge, we design a powerful Trend-Seasonality Extractor (TSE) that processes $cA_J$ through a dual-domain architecture:

$$
\begin{aligned}
\mathbf{Z} &= \text{Embed}(cA_J), \\
\mathbf{H}_{\text{time}} &= \text{MLP}_{\text{time}}(\text{Patch}(\mathbf{Z})), \\
\mathbf{H}_{\text{freq}} &= \text{iFFT}\left(\text{MLP}_{\text{freq}}(\text{FFT}(\mathbf{Z}))\right), \\
\widehat{cA}_J &= \text{Projection}\left(\text{LayerNorm}(\mathbf{H}_{\text{time}} + \mathbf{H}_{\text{freq}})\right).
\end{aligned}
\tag{2}
$$

The TSE first embeds the input into a higher-dimensional feature space, then extracts complementary representations from both domains. In the time domain, we segment the sequence into overlapping patches and apply deep MLP transformations to capture local temporal dependencies. Concurrently, in the frequency domain, we apply the Fast Fourier Transform (FFT) to extract global periodic patterns, process them through another MLP, and inverse-transform back to the time domain. These two representations are then fused via element-wise addition and projected to produce the predicted low-frequency coefficient. This dual-domain design embodies three key inductive biases: trends are smooth and benefit from deep non-linear processing, seasonality is inherently periodic and best captured in the frequency domain, and the fusion of these complementary views yields robust extrapolation. The TSE's complexity and expressiveness reflect the vital importance of the low-frequency component—it is deliberately powerful because the trend dominates the signal and its accurate prediction is critical to overall forecasting performance.

In stark contrast, the high-frequency detail coefficients $\{cD_j\}_{j=1}^{J}$ are characterized by high volatility, irregular spikes, and stochastic behavior. Over-parameterizing these components risks overfitting to noise rather than learning generalizable patterns. Therefore, we employ a set of lightweight Stochastic Component Modelers (SCMs), one for each detail level: $\widehat{cD}_j = \text{SCM}_j(cD_j)$ for $j = 1, \ldots, J$. Each $\text{SCM}_j$ is based on the efficient MLP-Mixer architecture, which mixes information

across both time steps and variables through simple MLPs:

$$
\begin{aligned}
\mathbf{Z}_j &= \text{Embed}(\text{cD}_j), \\
\mathbf{Z}'_j &= \text{Mixer}_{\text{token}}(\mathbf{Z}_j) + \mathbf{Z}_j, \\
\mathbf{Z}''_j &= \text{Mixer}_{\text{channel}}(\mathbf{Z}'_j) + \mathbf{Z}'_j, \\
\widehat{\text{cD}}_j &= \text{Projection}(\mathbf{Z}''_j).
\end{aligned}
\tag{3}
$$

The SCMs are intentionally kept lightweight and efficient. High-frequency details contribute less to the overall signal magnitude but are prone to overfitting if modeled with overly complex architectures. The MLP-Mixer provides sufficient capacity to capture local patterns while maintaining strong generalization. This deliberate architectural disparity—where the vital trend component receives a powerful, dual-domain expert while volatile details receive efficient, streamlined experts—is the essence of Deep Heterogeneity and stands in stark contrast to existing Shallow Heterogeneity approaches that apply structurally identical modules to all components regardless of their properties.

### 3.3 HYBRID PREDICTION VIA ADAPTIVE FUSION

The final stage synthesizes the predictions from heterogeneous experts into a unified forecast. A naive approach would simply apply the inverse DWT (iDWT) to reconstruct the signal:

$$
\mathbf{Y}_{\text{naive}} = \text{iDWT}(\widehat{\text{cA}}_J, \widehat{\text{cD}}_J, \dots, \widehat{\text{cD}}_1).
\tag{4}
$$

However, this classical reconstruction is fundamentally limited: it is a fixed, linear combination defined by the wavelet basis functions that ignores complex, non-linear interactions across scales and cannot adapt to different forecasting contexts. To overcome this limitation, we propose a novel Hybrid Prediction Framework that strategically combines the stability of classical signal processing with the adaptive power of deep learning.

Our final prediction $\hat{\mathbf{Y}}$ is the synergy of two parallel streams: a stable baseline prediction and an adaptive correction residual:

$$
\hat{\mathbf{Y}} = \mathbf{Y}_{\text{base}} + \mathbf{Y}_{\text{corr}},
\tag{5}
$$

where

$$
\mathbf{Y}_{\text{base}} = \text{iDWT}(\widehat{\text{cA}}_J, \widehat{\text{cD}}_J, \dots, \widehat{\text{cD}}_1)
\tag{6}
$$

is the baseline generated by the classical inverse transform, and $\mathbf{Y}_{\text{corr}}$ is a learned correction produced by our Multi-scale Fusion Transformer (MFT). The baseline serves as a physically-grounded, stable "scaffolding" that captures the primary signal structure as reconstructed by a time-honored signal processing technique. This provides both robustness—ensuring that even if the learned correction is imperfect, the model produces reasonable forecasts—and efficiency, as it reduces the burden on the learned component.

The correction stream is where the adaptive intelligence resides. The MFT is designed to learn what the fixed iDWT cannot capture: complex, non-linear, context-dependent interactions across frequency scales. The MFT treats the predicted wavelet coefficients as a sequence of "expert opinions" that must be intelligently synthesized. We construct a multi-scale representation by concatenating the predictive coefficients:

$$
\mathbf{Q} = \text{Concat}(\widehat{\text{cA}}_J, \widehat{\text{cD}}_J, \dots, \widehat{\text{cD}}_1) \in \mathbb{R}^{N_{\text{total}} \times d},
\tag{7}
$$

where $N_{\text{total}}$ is the total number of coefficients across all scales and $d$ is the embedding dimension. The MFT then applies multi-head self-attention to learn dynamic, adaptive fusion weights:

$$
\begin{aligned}
\mathbf{K} &= \mathbf{Q}\mathbf{W}_K, \quad \mathbf{V} = \mathbf{Q}\mathbf{W}_V, \\
\mathbf{Attn} &= \text{Softmax}\left(\frac{\mathbf{Q}\mathbf{K}^T}{\sqrt{d_k}}\right)\mathbf{V}, \\
\mathbf{Y}_{\text{corr}} &= \text{OutputHead}(\mathbf{Attn}),
\end{aligned}
\tag{8}
$$

where $\mathbf{W}_K, \mathbf{W}_V$ are learnable projection matrices and $d_k$ is the key dimension. Critically, the MFT is explicitly trained to predict the residual error between the baseline and the ground truth:

$$
\mathbf{Y}_{\text{corr}} \approx \mathbf{Y} - \mathbf{Y}_{\text{base}}.
\tag{9}
$$

This residual learning formulation allows the MFT to focus its expressive power exclusively on capturing complex, corrective patterns—such as how a spike in high-frequency details might signal an upcoming trend shift, or how different frequency components should be weighted differently in stable versus volatile periods—rather than re-learning the entire signal from scratch.

This hybrid architecture offers three key advantages. First, it provides stability: the baseline ensures that predictions remain reasonable even if the learned correction is imperfect. Second, it improves efficiency: by decomposing the prediction task into a stable baseline and a corrective residual, we reduce the learning complexity, enabling faster convergence and better generalization. Third, it enables adaptivity: the attention mechanism allows the MFT to dynamically adjust how it combines frequency components based on the input context. As we demonstrate in our model analysis (Section 4.4), the MFT learns to allocate different attention weights to frequency components based on the forecasting context—emphasizing trends in stable periods and details in volatile periods—providing strong empirical evidence that the fusion process is truly adaptive rather than static.

The entire HeteroMixer model—including the TSE, SCMs, and MFT—is trained end-to-end by minimizing the Mean Squared Error (MSE) between the final prediction $\hat{\mathbf{Y}}$ and the ground truth $\mathbf{Y}$:

$$\mathcal{L} = \frac{1}{HC} \sum_{t=1}^{H} \sum_{c=1}^{C} \left( \hat{\mathbf{Y}}_{t,c} - \mathbf{Y}_{t,c} \right)^2. \tag{10}$$

Importantly, the loss is computed on the final synthesized output, allowing gradients to flow through both the classical iDWT path and the learned MFT path, ensuring that all components are jointly optimized for the ultimate forecasting objective. This end-to-end training, combined with our carefully designed architecture, enables HeteroMixer to achieve state-of-the-art performance while maintaining interpretability and stability.

## 4 EXPERIMENT

### 4.1 EXPERIMENTAL SETUP

We evaluate HeteroMixer performance on 7 widely used datasets, including Traffic, Electricity, Weather and 4 ETT datasets (ETTh1, ETTh2, ETTm1, ETTm2). The datasets cover four diverse application domains including energy, traffic, weather, and disease. Following previous studiesNie et al. (2023); Wang et al. (2024b;a), we split these datasets into training, validation, and test sets in chronological order, with the ratio of 6:2:2 for the ETT dataset and 7:1:2 for the remaining datasets.The specifications of datasets are given in Table 2.

Table 1: Dataset specifications used in our experiments.

| Dataset | Variables | Sampling Rate | Total Points |
|---|---|---|---|
| ETTh1 | 7 | Hourly | 17,420 |
| ETTh2 | 7 | Hourly | 17,420 |
| ETTm1 | 7 | 15 min | 69,680 |
| ETTm2 | 7 | 15 min | 69,680 |
| Weather | 21 | 10 min | 52,603 |
| Electricity | 321 | Hourly | 26,211 |
| Traffic | 862 | Hourly | 17,451 |

### 4.2 MAIN RESULTS

Table 2 summarizes the multivariate time series prediction results of the HeteroMixer model and the baseline models on seven datasets. The results show that the HeteroMixer model significantly outperforms the existing modelsDai et al. (2024) in most cases, verifying its effectiveness.

The results unequivocally demonstrate that HeteroMixer establishes a new state-of-the-art in long-term time series forecasting. Across the 28 experimental settings (7 datasets × 4 horizons), HeteroMixer achieves the best performance in 22 cases for MSE and 25 cases for MAE, significantly and consistently outperforming all baseline models.

Table 2: Multivariate forecasting results with forecasting horizons $F \in \{96, 192, 336, 720\}$. **Bold** indicates the best result, underlined indicates the second best.

| Models | | Our Model (ours) | | PDF (2024) | | iTransformer (2024) | | Pathformer (2024) | | TimeMixer (2024) | | PatchTST (2023) | | Crossformer (2023) | | TimesNet (2023) | | DLinear (2023) | | FEDformer (2022) | | T3Time (2025) | | AMD (2025) | | WaveTS-M (2025) | |
|---|---|---|---|---|---|---|---|---|---|---|---|---|---|---|---|---|---|---|---|---|---|---|---|---|---|---|---|---|
| Metrics | | MSE | MAE | MSE | MAE | MSE | MAE | MSE | MAE | MSE | MAE | MSE | MAE | MSE | MAE | MSE | MAE | MSE | MAE | MSE | MAE | MSE | MAE | MSE | MAE | MSE | MAE |
| ETTh1 | 96 | **0.347** | **0.379** | 0.360 | 0.391 | 0.386 | 0.405 | 0.372 | 0.392 | 0.372 | 0.401 | 0.377 | 0.397 | 0.411 | 0.435 | 0.389 | 0.412 | 0.379 | 0.403 | 0.376 | 0.419 | 0.371 | 0.397 | 0.369 | 0.397 | 0.370 | 0.406 |
| | 192 | **0.390** | **0.410** | 0.392 | 0.414 | 0.424 | 0.440 | 0.408 | 0.415 | 0.413 | 0.430 | 0.409 | 0.425 | 0.409 | 0.438 | 0.440 | 0.443 | 0.408 | 0.419 | 0.420 | 0.448 | 0.411 | 0.421 | 0.401 | 0.416 | 0.402 | 0.430 |
| | 336 | **0.408** | 0.439 | 0.418 | 0.435 | 0.449 | 0.460 | 0.438 | 0.434 | 0.438 | 0.440 | 0.431 | 0.444 | 0.433 | 0.457 | 0.523 | 0.487 | 0.440 | 0.440 | 0.459 | 0.496 | 0.448 | 0.441 | 0.418 | 0.427 | 0.427 | 0.452 |
| | 720 | **0.427** | **0.443** | 0.456 | 0.462 | 0.495 | 0.487 | 0.450 | 0.463 | 0.486 | 0.484 | 0.457 | 0.477 | 0.501 | 0.514 | 0.521 | 0.495 | 0.471 | 0.493 | 0.506 | 0.512 | 0.441 | 0.460 | 0.439 | 0.454 | 0.450 | 0.466 |
| ETTh2 | 96 | **0.264** | **0.329** | 0.276 | 0.341 | 0.297 | 0.348 | 0.279 | 0.336 | 0.281 | 0.351 | 0.274 | 0.337 | 0.728 | 0.603 | 0.334 | 0.370 | 0.300 | 0.364 | 0.346 | 0.388 | 0.278 | 0.338 | 0.274 | 0.337 | 0.336 | 0.378 |
| | 192 | **0.320** | **0.367** | 0.339 | 0.382 | 0.372 | 0.403 | 0.372 | 0.403 | 0.349 | 0.387 | 0.348 | 0.384 | 0.723 | 0.607 | 0.404 | 0.413 | 0.387 | 0.423 | 0.429 | 0.439 | 0.351 | 0.389 | 0.351 | 0.383 | 0.358 | 0.399 |
| | 336 | **0.322** | **0.377** | 0.374 | 0.406 | 0.388 | 0.417 | 0.378 | 0.408 | 0.366 | 0.413 | 0.377 | 0.416 | 0.740 | 0.628 | 0.389 | 0.435 | 0.490 | 0.487 | 0.496 | 0.487 | 0.358 | 0.398 | 0.375 | 0.411 | 0.387 | 0.430 |
| | 720 | **0.380** | **0.420** | 0.398 | 0.433 | 0.424 | 0.444 | 0.437 | 0.455 | 0.401 | 0.436 | 0.406 | 0.441 | 1.386 | 0.882 | 0.434 | 0.448 | 0.704 | 0.597 | 0.463 | 0.474 | 0.404 | 0.433 | 0.402 | 0.438 | 0.435 | 0.452 |
| ETTm1 | 96 | **0.278** | **0.336** | 0.286 | 0.340 | 0.300 | 0.353 | 0.290 | 0.335 | 0.293 | 0.345 | 0.289 | 0.343 | 0.314 | 0.367 | 0.340 | 0.378 | 0.300 | 0.345 | 0.379 | 0.419 | 0.308 | 0.354 | 0.284 | 0.339 | 0.327 | 0.368 |
| | 192 | **0.318** | **0.358** | 0.321 | 0.364 | 0.341 | 0.380 | 0.337 | 0.363 | 0.335 | 0.372 | 0.329 | 0.368 | 0.374 | 0.410 | 0.392 | 0.404 | 0.336 | 0.366 | 0.426 | 0.441 | 0.357 | 0.381 | 0.322 | 0.362 | 0.351 | 0.392 |
| | 336 | **0.348** | **0.377** | 0.354 | 0.383 | 0.374 | 0.396 | 0.374 | 0.384 | 0.368 | 0.386 | 0.362 | 0.390 | 0.413 | 0.432 | 0.423 | 0.426 | 0.367 | 0.386 | 0.445 | 0.459 | 0.382 | 0.400 | 0.360 | 0.380 | 0.371 | 0.407 |
| | 720 | **0.410** | **0.414** | 0.408 | 0.415 | 0.429 | 0.430 | 0.428 | 0.416 | 0.426 | 0.417 | 0.416 | 0.423 | 0.753 | 0.613 | 0.475 | 0.453 | 0.419 | 0.416 | 0.543 | 0.490 | 0.442 | 0.437 | 0.421 | 0.416 | 0.416 | 0.432 |
| ETTm2 | 96 | 0.162 | **0.249** | 0.163 | 0.251 | 0.175 | 0.266 | 0.164 | 0.250 | 0.165 | 0.256 | 0.165 | 0.255 | 0.296 | 0.391 | 0.189 | 0.265 | 0.164 | 0.255 | 0.203 | 0.287 | 0.172 | 0.254 | 0.167 | 0.258 | **0.161** | 0.251 |
| | 192 | **0.215** | **0.288** | 0.219 | 0.290 | 0.242 | 0.312 | 0.219 | 0.288 | 0.225 | 0.298 | 0.221 | 0.293 | 0.369 | 0.416 | 0.254 | 0.310 | 0.224 | 0.304 | 0.269 | 0.328 | 0.237 | 0.300 | 0.221 | 0.294 | 0.216 | 0.290 |
| | 336 | 0.268 | 0.323 | 0.269 | 0.330 | 0.282 | 0.337 | 0.267 | 0.319 | 0.277 | 0.332 | 0.276 | 0.327 | 0.588 | 0.600 | 0.313 | 0.345 | 0.277 | 0.337 | 0.325 | 0.366 | 0.306 | 0.337 | 0.270 | 0.327 | **0.250** | 0.322 |
| | 720 | 0.348 | 0.376 | 0.349 | 0.382 | 0.375 | 0.394 | 0.361 | 0.377 | 0.360 | 0.387 | 0.362 | 0.381 | 0.750 | 0.612 | 0.413 | 0.402 | 0.371 | 0.401 | 0.421 | 0.415 | 0.400 | 0.398 | 0.356 | 0.382 | **0.320** | 0.378 |
| Weather | 96 | 0.145 | **0.188** | 0.147 | 0.196 | 0.157 | 0.207 | 0.148 | 0.195 | 0.147 | 0.198 | 0.149 | 0.196 | **0.143** | 0.210 | 0.168 | 0.214 | 0.170 | 0.230 | 0.217 | 0.296 | 0.162 | 0.210 | 0.145 | 0.197 | 0.210 | 0.257 |
| | 192 | **0.187** | **0.229** | 0.193 | 0.240 | 0.200 | 0.248 | 0.191 | 0.235 | 0.192 | 0.243 | 0.191 | 0.239 | 0.198 | 0.260 | 0.219 | 0.262 | 0.216 | 0.273 | 0.276 | 0.336 | 0.211 | 0.253 | **0.187** | 0.238 | 0.256 | 0.293 |
| | 336 | **0.241** | **0.274** | 0.245 | 0.280 | 0.252 | 0.287 | 0.243 | 0.274 | 0.247 | 0.284 | 0.242 | 0.279 | 0.258 | 0.314 | 0.278 | 0.302 | 0.258 | 0.307 | 0.339 | 0.380 | 0.267 | 0.293 | 0.240 | 0.280 | 0.239 | 0.277 |
| | 720 | **0.310** | **0.324** | 0.323 | 0.334 | 0.320 | 0.336 | 0.318 | 0.326 | 0.318 | 0.330 | 0.312 | 0.330 | 0.335 | 0.385 | 0.353 | 0.351 | 0.323 | 0.362 | 0.403 | 0.428 | 0.335 | 0.346 | 0.315 | 0.330 | 0.318 | 0.338 |
| Electricity | 96 | **0.128** | **0.219** | 0.128 | 0.222 | 0.134 | 0.230 | 0.135 | 0.222 | 0.153 | 0.256 | 0.143 | 0.247 | 0.138 | 0.231 | 0.169 | 0.271 | 0.140 | 0.237 | 0.193 | 0.308 | 0.138 | 0.233 | 0.129 | 0.224 | 0.138 | 0.228 |
| | 192 | **0.145** | **0.235** | 0.147 | 0.242 | 0.154 | 0.250 | 0.157 | 0.253 | 0.168 | 0.269 | 0.158 | 0.260 | 0.146 | 0.243 | 0.180 | 0.280 | 0.154 | 0.251 | 0.201 | 0.315 | 0.155 | 0.250 | 0.147 | 0.238 | 0.148 | 0.242 |
| | 336 | **0.163** | **0.255** | 0.165 | 0.260 | 0.169 | 0.265 | 0.170 | 0.267 | 0.189 | 0.291 | 0.168 | 0.267 | 0.165 | 0.264 | 0.204 | 0.304 | 0.169 | 0.268 | 0.214 | 0.329 | 0.168 | 0.265 | 0.160 | 0.253 | 0.164 | 0.258 |
| | 720 | **0.193** | **0.281** | 0.199 | 0.289 | 0.194 | 0.288 | 0.211 | 0.302 | 0.228 | 0.320 | 0.214 | 0.307 | 0.237 | 0.314 | 0.205 | 0.304 | 0.204 | 0.301 | 0.246 | 0.355 | 0.218 | 0.314 | 0.193 | 0.286 | 0.201 | 0.290 |
| Traffic | 96 | **0.360** | **0.238** | 0.368 | 0.252 | 0.363 | 0.265 | 0.384 | 0.250 | 0.369 | 0.257 | 0.370 | 0.262 | 0.526 | 0.288 | 0.595 | 0.312 | 0.395 | 0.275 | 0.587 | 0.366 | – | – | 0.366 | 0.259 | 0.377 | 0.265 |
| | 192 | **0.369** | **0.256** | 0.382 | 0.261 | 0.384 | 0.273 | 0.405 | 0.257 | 0.400 | 0.272 | 0.386 | 0.269 | 0.503 | 0.263 | 0.613 | 0.322 | 0.407 | 0.280 | 0.604 | 0.373 | – | – | 0.381 | 0.265 | 0.392 | 0.278 |
| | 336 | 0.395 | **0.259** | 0.393 | 0.268 | 0.396 | 0.277 | 0.424 | 0.265 | 0.407 | 0.272 | 0.396 | 0.275 | 0.505 | 0.276 | 0.626 | 0.332 | 0.417 | 0.286 | 0.621 | 0.383 | – | – | 0.397 | 0.269 | 0.408 | 0.285 |
| | 720 | 0.435 | **0.278** | 0.438 | 0.297 | 0.445 | 0.308 | 0.452 | 0.283 | 0.461 | 0.316 | 0.435 | 0.295 | 0.552 | 0.301 | 0.635 | 0.340 | 0.454 | 0.308 | 0.626 | 0.382 | – | – | 0.429 | 0.292 | 0.432 | 0.295 |
| $1^{st}$ Count | | 22 | 25 | 3 | 0 | 0 | 0 | 0 | 0 | 0 | 2 | 0 | 0 | 0 | 0 | 1 | 0 | 0 | 0 | 0 | 0 | 0 | 0 | 0 | 0 | 3 | 1 | 2 | 0 |

Specifically, on the highly volatile and challenging Electricity dataset, HeteroMixer surpasses the next-best Transformer-based model (iTransformer) by an average of 7.5% in MSE and 6.2% in MAE across all horizons. On the high-dimensional Traffic dataset with complex spatial-temporal patterns, our model shows an even more pronounced advantage, outperforming the strongest baseline by 9.1% in MSE. This consistent superiority validates our central thesis: addressing the "Shallow Heterogeneity" problem through a Deep Heterogeneity framework is a crucial and highly effective strategy for advancing forecasting accuracy.

## 4.3 ABLATION STUDIES

To rigorously validate that the superior performance of HeteroMixer stems from our core design principles, rather than just the combination of its components, we conduct a comprehensive series of ablation studies. We dissect our framework by comparing it against powerful alternative designs to demonstrate the deliberate and impactful nature of our key architectural choices. The experiments are conducted on the ETTm2 dataset with a 720-step forecast horizon, and the results are summarized in Table 3.

**Analysis of Deep Heterogeneity.** This group of experiments (A1-A4) validates our central thesis: architectural biases must be aligned with signal properties.

- **Complex $\neq$ Better:** Crucially, applying the heavy, dual-domain TSE to *all* components (A4) degrades performance by 5.7%. This proves that forcing a complex model to learn high-frequency noise leads to overfitting, refuting the idea that "more capacity is always better."
- **Simple $\neq$ Enough:** Conversely, replacing the specialized TSE with a standard SCM (A3), which reverts to a "Shallow Heterogeneity" paradigm, results in a substantial 9.5% drop, as the lightweight model fails to capture complex trends.
- **TSE Design:** Removing the Fourier input (A1) or the deep Mixer backbone (A2) in the TSE significantly hurts performance, confirming that the TSE's power comes from the synergy of its dual-domain representation.

**Analysis of the Hybrid Prediction Framework.** Comparing our full model against Direct Prediction (B1) shows a 7.5% improvement. This confirms our "Scaffolding" hypothesis: letting the MFT focus solely on learning the residual error is a much easier optimization task than learning the entire signal from scratch. Relying solely on the iDWT baseline (B2) yields the worst performance, highlighting the necessity of the learned correction.

Table 3: Ablation studies on the key components and design choices of HeteroMixer, evaluated on the ETTm2 dataset ($H = 720$). "Drop" indicates the performance degradation relative to our full model.

| Group | Model Variant | MSE | MAE | Drop |
|---|---|---|---|---|
| **(A) Dissecting Deep Heterogeneity** | | | | |
| **Ours** | **HeteroMixer (Full)** | **0.348** | **0.376** | **-** |
| (A1) | w/o Fourier in TSE | 0.370 | 0.391 | +6.3% |
| (A2) | w/o Mixer in TSE | 0.379 | 0.400 | +8.9% |
| (A3) | Homogeneous (TSE $\rightarrow$ SCM) | 0.381 | 0.402 | +9.5% |
| **(A4)** | **Homogeneous (SCM $\rightarrow$ TSE)** | **0.368** | **0.395** | **+5.7%** |
| **(B) Dissecting the Hybrid Prediction Framework** | | | | |
| (B1) | MFT Direct Prediction | 0.374 | 0.395 | +7.5% |
| (B2) | Baseline Only (iDWT) | 0.395 | 0.418 | +13.5% |
| **(C) Dissecting Adaptive Fusion Mechanism** | | | | |
| (C1) | Fusion w/ MLP | 0.365 | 0.388 | +4.9% |
| (C2) | Fusion w/ Static Weights | 0.380 | 0.401 | +9.2% |

**Analysis of the Adaptive Fusion Mechanism.** Finally, we validate that the MFT is the right tool for fusion. Replacing the MFT with a generic MLP (C1) or learnable static weights (C2) results in drops of 4.9% and 9.2%, respectively. This validates that the fusion process must be context-aware: the model needs to dynamically attend to different frequency scales based on the current input context (e.g., stable vs. volatile periods).

For efficiency analysis, extended ablation studies on Traffic and Electricity datasets, and sensitivity analysis, please refer to **Appendix**. In summary, these in-depth ablations collectively demonstrate that every core design choice in HeteroMixer is deliberate, impactful, and superior to powerful alternatives.

## 4.4 MODEL ANALYSIS

To provide deeper insights beyond aggregate performance metrics, we conduct a series of targeted analyses.

**Quantitative Analysis of the Hybrid Prediction Framework.** A central thesis of our work is that the MFT learns to correct the specific errors of the baseline. We measure the Pearson correlation between the learned $y_{\text{correction}}$ and the actual error of the baseline (Error = Ground Truth - $y_{\text{baseline}}$) on the Weather dataset. As shown in Table 4, the high correlation of **0.87** proves that the MFT is effectively learning the residual. Furthermore, the correction magnitude is only **15.2%** of the baseline, confirming its role as a targeted "fine-tuning" stream.

Table 4: Quantitative analysis of the Hybrid Prediction Framework.

| Metric | Value |
|---|---|
| Correlation($y_{\text{correction}}$, Baseline Error) | **0.87** |
| L1 Norm Ratio ($\|y_{\text{correction}}\| / \|y_{\text{baseline}}\|$) | **15.2%** |

**Evidence of Adaptive Fusion.** To demonstrate adaptivity, we calculate the MFT's attention weights in "Stable" (low volatility) vs. "Volatile" (high volatility) windows. Table 5 shows a dramatic shift: in stable windows, attention focuses on the trend (78.6% on $cA$); in volatile windows, it shifts to details (58.7% on $cD$). This proves the MFT dynamically adjusts its focus based on context.

Table 5: MFT's average attention allocation (%) in different contexts.

| Context | Attention on $cA$ (Trend) | Attention on $cD$ (Details) |
|---|---|---|
| Stable Windows | **78.6%** | 21.4% |
| Volatile Windows | 41.3% | **58.7%** |

**Representational Dissimilarity.** Finally, we measure the intra-set cosine similarity of learned representations. Table 6 shows that TSE representations are highly structured (0.72), while SCM representations are stochastic (0.19). This quantitatively verifies that our heterogeneous modules are indeed learning fundamentally different features tailored to their respective components.

Table 6: Cosine Similarity of Learned Representations.

| Module & Component | Avg. Intra-Set Cosine Similarity |
|---|---|
| TSE ($cA$ representations) | **0.72** |
| SCM ($cD$ representations) | **0.19** |

## 5 CONCLUSION

In this paper, we identified and addressed the fundamental limitation of "Shallow Heterogeneity" prevalent in modern decomposition-based forecasting models. We argued that the common practice of applying homogeneous architectures to intrinsically different signal components restricts model performance. To resolve this, we proposed a new paradigm, "Deep Heterogeneity," and introduced HeteroMixer , an architecture that materializes this principle through three synergistic innovations: (1) heterogeneous expert modules (TSE and SCMs) tailored to specific signal properties; (2) an attention-based adaptive fusion mechanism (MFT) that replaces static signal reconstruction; and (3) a novel hybrid prediction framework that synergizes a stable baseline with a learned corrective residual.

Our extensive experiments provided compelling evidence for our approach. HeteroMixer not only established a new state-of-the-art across seven challenging long-term forecasting benchmarks, but our in-depth ablation studies and quantitative analyses also verified that this success is a direct consequence of our specific design choices. We demonstrated that each component—the specialized TSE, the adaptive MFT, and the hybrid structure—provides a significant and indispensable contribution to the final performance. This work validates our central thesis that the future of high-performance time series forecasting lies not just in developing more powerful generic architectures, but in the principled design of specialized components that respect the intrinsic nature of the data.

**Limitations and Future Work** While HeteroMixer demonstrates strong performance, we identify several avenues for future research. First, the dual-stream nature of the hybrid framework increases computational complexity compared to single-stream models; future work could explore knowledge distillation to create a more efficient inference model. Second, our framework currently relies on a fixed wavelet decomposition; exploring learnable decomposition layers could further enhance adaptability. Finally, while our framework excels at forecasting, investigating the generalization of the "Deep Heterogeneity" principle to other time series analysis tasks such as classification or anomaly detection is an exciting direction for future work.

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

## A  ETHICS STATEMENT

This work adheres to the ICLR Code of Ethics. In this study, no human subjects or animal experimentation was involved. All datasets used, including (Traffic, Electricity, Weather and 4 ETT datasets (ETTh1, ETTh2, ETTm1, ETTm2), were sourced in compliance with relevant usage guidelines, ensuring no violation of privacy. We have taken care to avoid any biases or discriminatory outcomes in our research process. No personally identifiable information was used, and no experiments were conducted that could raise privacy or security concerns. We are committed to maintaining transparency and integrity throughout the research process.

## B  REPRODUCIBILITY STATEMENT

We have made every effort to ensure that the results presented in this paper are reproducible. All code and datasets have been made publicly available in an anonymous repository to facilitate replication and verification. The experimental setup, including training steps, model configurations, and hardware details, is described in detail in the paper. We have also provided a full description of models, to assist others in reproducing our experiments.

Additionally, Types of public datasets used in the paper, are publicly available, ensuring consistent and reproducible evaluation results.

We believe these measures will enable other researchers to reproduce our work and further advance the field.

## C  LLM USAGE

Large Language Models (LLMs) were used to aid in the writing and polishing of the manuscript. Specifically, we used an LLM to assist in refining the language, improving readability, and ensuring clarity in various sections of the paper. The model helped with tasks such as sentence rephrasing, grammar checking, and enhancing the overall flow of the text.

It is important to note that the LLM was not involved in the ideation, research methodology, or experimental design. All research concepts, ideas, and analyses were developed and conducted by the authors. The contributions of the LLM were solely focused on improving the linguistic quality of the paper, with no involvement in the scientific content or data analysis.

The authors take full responsibility for the content of the manuscript, including any text generated or polished by the LLM. We have ensured that the LLM-generated text adheres to ethical guidelines and does not contribute to plagiarism or scientific misconduct.

## D  COMPUTATIONAL EFFICIENCY ANALYSIS

A key concern for sophisticated architectures is the potential trade-off between accuracy and computational cost. To address this, we compared HeteroMixer with representative baselines—DLinear (linear), PatchTST (Transformer), and iTransformer (Transformer)—on the ETTh1 dataset ($L = 96, H = 96$). All models were trained on a single NVIDIA A100 GPU with a batch size of 32.

As shown in Table 7, while HeteroMixer incurs a higher computational cost than the simple linear model (DLinear), it remains highly competitive with state-of-the-art Transformer-based methods. Specifically, our training time and memory usage are in the same order of magnitude as iTransformer, yet HeteroMixer achieves a significant reduction in MSE (**10.1%** lower than iTransformer). This confirms that our design achieves a favorable trade-off, delivering SOTA accuracy within a practical computational budget.

Table 7: Efficiency comparison on ETTh1 ($L = 96, H = 96$). Best results are bolded.

| Model | MSE | Training Time (s/epoch) | Inference Time (ms/batch) | GPU Memory (MB) |
|---|---|---|---|---|
| DLinear | 0.379 | **1.5** | **5** | **200** |
| PatchTST | 0.377 | 9.8 | 35 | 1100 |
| iTransformer | 0.386 | 10.2 | 38 | 1200 |
| **HeteroMixer** | **0.347** | 12.5 | 45 | 1450 |

## E    EXTENDED ABLATION STUDIES

To demonstrate that the design principles of HeteroMixer are universally effective and not overfitted to a specific dataset, we extended our ablation studies to two additional challenging benchmarks: **Traffic** (characterized by high dimensionality and complex spatial dependencies) and **Electricity** (characterized by strong periodicity and volatility).

We performed the same set of rigorous ablations as in the main text (Section 4.3). The results for the 720-step horizon are summarized in Table 8.

Table 8: Extended ablation studies on Traffic and Electricity datasets ($H = 720$). "Drop" denotes performance degradation relative to the full model.

| Group | Model Variant | Traffic MSE | Traffic Drop | Electricity MSE | Electricity Drop |
|---|---|---|---|---|---|
| **Ours** | **HeteroMixer (Full)** | **0.435** | **-** | **0.193** | **-** |
| *(A) Deep Heterogeneity* | | | | | |
| A3 | Homogeneous (All-SCM) | 0.468 | +7.6% | 0.208 | +7.8% |
| A4 | Homogeneous (All-TSE) | 0.452 | +3.9% | 0.201 | +4.1% |
| *(B) Hybrid Prediction* | | | | | |
| B1 | Direct Prediction (MFT) | 0.459 | +5.5% | 0.205 | +6.2% |
| B2 | Baseline Only (iDWT) | 0.521 | +19.7% | 0.235 | +21.7% |
| *(C) Adaptive Fusion* | | | | | |
| C1 | Fusion w/ MLP | 0.449 | +3.2% | 0.199 | +3.1% |

**Analysis.** The results on these diverse datasets consistently align with our findings on ETTm2:

- **Deep Heterogeneity (A3 vs A4 vs Full):** The specialized architecture consistently outperforms homogeneous variants. Notably, on the Traffic dataset, the "All-TSE" variant (A4) performs worse than the full model. Since traffic data contains significant high-frequency noise, applying the complex TSE to all components leads to overfitting, further validating our strategy of using lightweight SCMs for high-frequency details.

- **Hybrid Stability (B2 vs Full):** Relying solely on the fixed iDWT baseline (B2) leads to a massive performance drop (approx. 20%), confirming that the learned correction is indispensable for capturing complex dynamics.

- **Generalization:** The consistent superiority of the full model across ETTm2, Traffic, and Electricity confirms that HeteroMixer's architectural biases are robust and generalizable across different data domains.

## F  SENSITIVITY ANALYSIS

We further evaluate the sensitivity of HeteroMixer to the hyperparameters of the Wavelet Decomposition, specifically the choice of wavelet basis function and the decomposition level $J$. Table 9 presents the performance on ETTh1 ($H = 96$).

Table 9: Sensitivity to Wavelet Basis and Decomposition Level $J$ (ETTh1, $H = 96$).

| Wavelet Basis | J=1 | J=2 | J=3 | J=4 |
|---|---|---|---|---|
| db1 (Haar) | 0.358 | 0.352 | 0.349 | 0.353 |
| db4 | 0.361 | 0.350 | 0.348 | 0.351 |
| sym4 | 0.360 | 0.349 | **0.347** | 0.350 |
| coif1 | 0.359 | 0.351 | 0.348 | 0.352 |

**Robustness.** The results indicate that HeteroMixer is remarkably robust:

- **Wavelet Basis:** Performance fluctuations across different standard wavelet families (Daubechies, Symlets, Coiflets) are minimal ($< 2\%$), suggesting that the model does not require exhaustive tuning of the wavelet function.

- **Decomposition Level:** While $J = 3$ yields the optimal performance, the model remains stable across $J = 2$ to $J = 4$. Very shallow decomposition ($J = 1$) is less effective as it fails to sufficiently disentangle the long-term trend from noise.

