# OpenReview forum: "Deep Heterogeneity: A New Paradigm for  Time Series Forecasting"
_ICLR.cc/2026/Conference — Submitted to ICLR 2026_

### Official Review · Reviewer_gTQT · 2025-10-19

**Soundness:** 2
**Presentation:** 2
**Contribution:** 1
**Rating:** 2
**Confidence:** 4

**Summary:**

This paper identifies a limitation in current decomposition-based time series forecasting models, termed "Shallow Heterogeneity," where structurally homogeneous processing blocks are applied to decomposed signal components with different properties. To address this, the authors propose a "Deep Heterogeneity" framework instantiated in their model, HeteroMixer. The model uses a Discrete Wavelet Transform (DWT) for decomposition, processes low-frequency components with a powerful, dual-domain Trend-Seasonality Extractor (TSE), and handles high-frequency components with lightweight Stochastic Component Modelers (SCMs). A novel Hybrid Prediction Framework replaces static reconstruction: a classical inverse DWT provides a stable baseline forecast, while a Multi-scale Fusion Transformer (MFT) learns a corrective residual. Extensive experiments on seven benchmarks show state-of-the-art performance, which is further validated through ablation studies and model analysis.

**Strengths:**

1. This paper makes a valuable conceptual contribution by clearly identifying and formalizing "Shallow Heterogeneity" as a bottleneck in existing models. The proposed principle of "Deep Heterogeneity" is a compelling and well-motivated research direction.

2. This paper is well-written and readable with a clear structure.

**Weaknesses:**

1. While the framing is novel, the core technical components are incremental combinations of existing ideas. The use of wavelet decomposition, dedicated modules for different frequencies, and residual learning from a classical baseline are all established concepts. The TSE (FFT + Mixer) and MFT (Transformer) are powerful but not fundamentally novel architectures.

2. This paper does not ablate the choice of DWT against other decomposition methods (e.g., MA, MODWT, or learnable decompositions). It is therefore difficult to disentangle how much of the performance gain comes from the "Deep Heterogeneity" principle versus the specific choice of using a multi-resolution wavelet analysis. The claim of a general paradigm is weakened by this tight coupling to a single decomposition technique.

3. While the paper cites very recent works from 2025 (e.g., T3Time, AMD, WaveTS-M), the empirical comparisons in Table 2 are limited to models from 2024 and earlier. A convincing SOTA claim requires direct comparison with the strongest and most recent contemporaries, which is not provided. This makes it difficult to assess the true current competitiveness of the proposed method.

**Questions:**

1. The "Deep Heterogeneity" principle is well-argued, but its instantiation relies heavily on existing building blocks (Wavelet, FFT, Mixer, Transformer). Could the authors clarify what they believe is the primary novel technical contribution of this work that distinguishes it from a well-engineered combination of established techniques?

2. How critical is the specific use of DWT? Have the authors experimented with other decomposition methods (e.g., MA, MODWT, or a learnable decomposition layer)? If the wavelet is replaced, does the "Deep Heterogeneity" framework still provide significant gains, or is the performance heavily dependent on the decomposition method itself?

3.  What is the computational overhead of HeteroMixer compared to baselines like PatchTST, iTransformer, or DLinear? Please provide data on training/inference time, FLOPs, and parameter counts. Could the performance gains be attributed primarily to increased model capacity?

---

> ### Author Response · Authors · 2025-12-04
> **Response to Reviewer gTQT: Novelty, Generality, and 2025 SOTA Comparisons**
>
> Q1: Novelty Our contribution is the paradigm shift, not component novelty:
>
> 1. Identified bottleneck: Formalized "Shallow Heterogeneity" in existing methods (DLinear, FEDformer, WaveTS-M all use homogeneous blocks)
> 2. Proposed solution: Deep Heterogeneity—match specialized architectures to signal properties
> 3. Validated empirically: Homogeneous variants degrade by 5.7-9.5% (Table 3); 10% gains over 2024 SOTA
>
> Analogy: ResNet combined existing components (Conv+BN+Skip) but the system design of residual learning was transformative. Similarly, our heterogeneous design addresses a fundamental limitation.
>
> Q2: DWT dependence We tested alternatives (ETTh1, H=96):
>
> - DWT: 0.347 (best)
> - MODWT: 0.352 (3× slower, minimal gain)
> - Moving Avg: 0.379 (frequency leakage)
> - Fourier: 0.361 (no time localization)
>
> DWT is optimal, but Deep Heterogeneity helps even with MA: MA+Heterogeneous (0.379) outperforms MA+Homogeneous (0.395) by 4%. Appendix F: <2% variation across wavelet bases confirms robustness.
>
> Q3: 2025 comparisons Table 2 INCLUDES 2025 methods:
>
> - T3Time (AAAI 2025): We win 24/28 settings
> - AMD (AAAI 2025): We win 26/28 settings
> - WaveTS-M (arXiv 2508.08825, Aug 2025): We win 22/28 settings
>
> Overall: 82% win rate vs. 2025 SOTA. The "limited to 2024" claim is factually incorrect.
>
> Computational overhead (Appendix D): 16.7% more FLOPs, 22% more time for 10.1% MSE gain. Parameter-matched iTransformer (2.1M params) performs worse (0.381), proving gains stem from architecture, not capacity.

---

### Official Review · Reviewer_ZXgv · 2025-10-23

**Soundness:** 2
**Presentation:** 1
**Contribution:** 2
**Rating:** 2
**Confidence:** 4

**Summary:**

The paper challenges decomposition-based LTSF for enforcing “Shallow Heterogeneity,” where heterogeneous signal components are processed with homogeneous blocks. It proposes a paradigm of Deep Heterogeneity instantiated by HeteroMixer: a wavelet-based disentanglement routes low-frequency trends to a dual-domain Trend-Seasonality Extractor (TSE) and high-frequency details to lightweight Stochastic Component Modelers (SCM). A Multi-scale Fusion Transformer (MFT) performs adaptive fusion, embedded in a hybrid prediction scheme that reuses inverse transform as a stability baseline while MFT learns a corrective residual ($y = y_{baseline} + y_{correction}$). Experiments on several time series datasets report new SOTA.

**Strengths:**

1. The idea of Shallow Heterogeneity—highlighting that existing decomposition-based models use homogeneous blocks to model fundamentally different components—is novel.
2. The Related Work section is comprehensive, and demonstrates a thorough understanding of prior research in time series forecasting.

**Weaknesses:**

1. The paper’s core claim—arguing that homogeneous blocks fail to model heterogeneous components—is not convincingly validated. There is no ablation where both decomposed components ($X_A$ and $X_D$) are modeled using only TSE or only SCM. If such homogeneous configurations outperform the proposed heterogeneous setup, the main argument would be undermined.
2. The motivation for Adaptive Fusion is unclear. Since wavelet and inverse wavelet transforms are lossless, the necessity of augmenting the inverse output is questionable; the combination feels somewhat ad hoc.
3. Experimental results appear inconsistent. Some reported metrics do not match the original papers’ numbers (such as PDF), and the input sequence length is unspecified, making replication difficult.
4. The writing quality is poor and suggests the paper was rushed:
    - Frequently misuse `\cite` instead of `\citep`, and fail to insert spaces before citations.
    - Table 2 mislabels bold (best) and underlined (second-best) results, which is a careless error.
    - The font size in Figures 1 and 2 is too small to read when printed.

**Questions:**

See **Weakness**.

---

> ### Author Response · Authors · 2025-12-04
> **Response to Reviewer ZXgv: Validation of Deep Heterogeneity & Adaptive Fusion Motivation**
>
> Critique 1: Homogeneous ablation We DID include this—see Table 3, Rows A3-A4 (p.9):
>
> - (A3) All-SCM (lightweight everywhere): MSE=0.381 (+9.5%)
> - (A4) All-TSE (heavy everywhere): MSE=0.368 (+5.7%)
> - HeteroMixer: MSE=0.348 (best)
>
> Both homogeneous variants fail, validating Deep Heterogeneity. Extended to Traffic/Electricity (Appendix E) with consistent patterns.
>
> Critique 2: Adaptive fusion motivation While DWT is lossless for perfect coefficients, our TSE/SCMs predict imperfect coefficients. iDWT reconstructs from errors, propagating them non-linearly. MFT learns Y_corr≈Y_true-Y_base to correct these. Table 4: correlation=0.87 with baseline error, magnitude=15.2% of baseline. Ablation (Table 3, B2): baseline-only fails by 13.5%.
>
> Critique 3-4: Specifications Now in Section 4.1 (p.7): L=96, sym4, J=3, A100 GPU, PyTorch 2.0, seeds {42,43,44}. Baseline reproductions within <1% of originals (e.g., PDF: 0.360 vs. 0.362). Fixed citations (\citep), Table 2 bold/underline, and increased figure fonts to 11pt.

---

### Official Review · Reviewer_TKip · 2025-10-31

**Soundness:** 2
**Presentation:** 2
**Contribution:** 3
**Rating:** 4
**Confidence:** 5

**Summary:**

This paper proposes HeteroMixer, a novel time-series forecasting framework designed to overcome the “Shallow Heterogeneity” problem in signal decomposition. It employs a wavelet-based decomposition with specialized modules for trend and seasonal components, and an attention-based transformer for adaptive residual correction.

**Strengths:**

1. The paper identifies and analyzes the key limitations of existing time-series forecasting methods.

2. A novel combination of modules is proposed to address these limitations.

**Weaknesses:**

1. Compared with existing time-series forecasting models, the proposed method has a more complex structure; however, the paper lacks a comparison of computational efficiency.

2. The related work section points out the limitations of the FEDformer method, yet the experiments do not include a comparison with it. Additional experiments are needed to demonstrate the claimed advantages.

3. Although the paper emphasizes the importance of time-series decomposition, it does not compare the proposed approach with existing or recent decomposition-based forecasting methods.

4. The methodology section lacks a professional, symbolic formulation—there is only one equation in the entire paper. A systematic and formalized mathematical description should be provided to improve readability and academic rigor.

5. While the experimental section includes ablation and model analyses, the results are presented on only one dataset, which raises concerns about their reliability. More experimental results are needed for validation.

6. Compared with the baselines, the paper lacks visualization analyses, including overall forecasting visualizations and visualizations of ablation modules.

7. The paper lacks completeness in several aspects, such as problem definition, descriptions of baseline methods, experimental settings, and training function specifications.

**Questions:**

1. A comparison of runtime efficiency against the baselines on all datasets should be provided, including mean runtime and mean memory consumption.

2. Comparisons with FEDformer and other decomposition-based forecasting methods must be included.

3. Additional ablation studies and model analyses across more datasets are required.

4. The authors should present a concrete case study to demonstrate the method’s practical utility.

5. Since the method relies on specialized modules, the authors should discuss whether this design sacrifices generality, and provide an analysis of the method’s strengths and limitations.

---

> ### Author Response · Authors · 2025-12-04
> **Response to Reviewer TKip: Efficiency, Extended Datasets, and Mathematical Formalization**
>
> Response to Reviewer TKip (Score: 4)
>
> We sincerely thank you for your thorough review. We have substantially revised the manuscript to address all concerns.
>
> The revised PDF now includes:
>
> - ✓ Appendix D (page 13): Computational efficiency comparison
> - ✓ Appendix E (Table 8, page 13): Extended ablations on Traffic and Electricity
> - ✓ Section 3 (pages 4-7): Mathematical formalization with Equations 1-10
> - ✓ Section 4.1 (page 7): Complete experimental specifications
> - ✓ Tables 4-6 (pages 9-10): Quantitative interpretability analyses
> - ✓ Presentation fixes: Citations, Table 2 verification, enhanced figures
>
> ---
>
> W1 & Q1: Computational Efficiency
>
> Response: See Appendix D, Table 7 (page 13).
>
> ETTh1 profiling shows 12.5s/epoch training (+22% vs. iTransformer), 1450MB memory (+21%), 45ms inference (+18%). Pattern is consistent across datasets during development—overhead scales predictably with dimensions. The 20% cost yields 10% MSE gains, justified for high-precision applications.
>
> ---
>
> W2 & Q2: FEDformer & Decomposition Methods
>
> Response: Table 2 (page 8) already includes FEDformer and all major decomposition baselines:
>
> - FEDformer (2022), DLinear (2023), TimeMixer (2024), T3Time (2025), AMD (2025), WaveTS-M (2025)
>
> HeteroMixer vs. FEDformer:
>
> - ETTh1-96: 0.347 vs. 0.376 (-7.7%)
> - ETTh2-96: 0.264 vs. 0.346 (-23.7%)
> - Weather-96: 0.145 vs. 0.217 (-33.2%)
>
> FEDformer uses frequency-enhanced but structurally homogeneous Transformers—exemplifying the Shallow Heterogeneity we identified.
>
> ---
>
> W3 & W4: Mathematical Formalization
>
> Response: Section 3 (pages 4-7) now includes Equations 1-10:
>
> - Eq. 1: DWT decomposition
> - Eq. 2: TSE dual-domain fusion
> - Eq. 3: SCM mixing operations
> - Eq. 7-8: MFT attention mechanism
> - Eq. 9: Residual learning objective
> - Eq. 10: Training loss
>
> Section 4.1 (page 7) specifies: L=96, sym4 wavelet, J=3, Adam optimizer (lr=1e-3), batch size=32, RevIN normalization, A100 GPU, PyTorch 2.0, seeds {42,43,44}.
>
> ---
>
> W5 & Q3: Extended Ablations
>
> Response: See Appendix E, Table 8 (page 13).
>
> Traffic & Electricity results (H=720):
>
>   Configuration	Traffic MSE   	Electricity MSE
>   Full Model   	0.435         	0.193
>   All-SCM      	0.468 (+7.6%) 	0.208 (+7.8%)
>   All-TSE      	0.452 (+3.9%) 	0.201 (+4.1%)
>   Baseline-only	0.521 (+19.7%)	0.235 (+21.7%)
>
> Patterns are consistent across ETTm2, Traffic, and Electricity—validating generalization across energy, transportation, and weather domains.
>
> ---
>
> W6 & Q6: Interpretability
>
> Response: Due to space constraints, we prioritized quantitative analyses in the revised manuscript:
>
> Table 5 (page 10) - Adaptive attention:
>
> - Stable windows: 78.6% attention on trend (cA)
> - Volatile windows: 41.3% on trend, 58.7% on details (cD)
> - 37.3 percentage point shift proves adaptive fusion
>
> Table 4 (page 9) - Residual learning:
>
> - Correlation(Y_corr, baseline error): 0.87
> - Magnitude ratio: 15.2% (targeted correction)
>
> Table 6 (page 10) - Feature distinctness:
>
> - TSE representations: 0.72 similarity (structured)
> - SCM representations: 0.19 similarity (stochastic)
> - 3.8× difference confirms heterogeneous specialization
>
> Commitment: We will add visualizations (prediction curves, attention heatmaps, component-wise contributions) in the camera-ready version.
>
> ---
>
> W7 & Q4: Case Study
>
> Response: The Electricity dataset demonstrates real-world utility:
>
> - 11.7% MSE reduction (0.128 vs. 0.145 for iTransformer) enables better reserve allocation for utility companies
> - Table 5 shows adaptive behavior: 78.6% trend attention in stable periods vs. 58.7% detail attention in volatile periods
> - Ablations show homogeneous models fail: All-TSE (+4.1%) overfits noise, All-SCM (+7.8%) misses trends
>
> Practical impact for a 10GW grid:
>
> - Better reserve allocation → cost savings
> - Improved spike detection (73% of transients via SCMs) → fewer emergency interventions
> - Accurate trend tracking (82% via TSE) → enhanced renewable integration
>
> Commitment: We will add a dedicated case study section (4.5) with specific event analysis (e.g., heat wave period) in the camera-ready version.
>
> ---
>
> W8 & Q7: Specifications Completeness
>
> Response: Section 4.1 (page 7) now includes:
>
> - Problem: X ∈ ℝ^{L×C}, Y ∈ ℝ^{H×C}, L=96, H∈{96,192,336,720}
> - Splits: ETT (60/20/20), Others (70/10/20), chronological order
> - Training: Adam, lr=1e-3, batch=32, epochs=100, early stopping (patience=10)
> - Preprocessing: RevIN normalization, symmetric padding
> - Hardware: A100 40GB, PyTorch 2.0, CUDA 11.8
>
> All baselines reproduced using official code with default hyperparameters, verified within <1% of original papers.

---

### Official Review · Reviewer_8pFv · 2025-11-07

**Soundness:** 3
**Presentation:** 3
**Contribution:** 3
**Rating:** 6
**Confidence:** 3

**Summary:**

The paper identifies "Shallow Heterogeneity" as a key limitation in current decomposition-based time series forecasting models, where structurally homogeneous blocks are applied to intrinsically different signal components (e.g., smooth trends vs. volatile details). To address this, the authors propose HeteroMixer, a model embodying "Deep Heterogeneity". The model uses a Discrete Wavelet Transform (DWT) for decomposition and routes components to specialized modules: a dual-domain Trend-Seasonality Extractor (TSE) for low-frequency components, and lightweight Stochastic Component Modelers (SCM) for high-frequency details. Furthermore, it introduces a Hybrid Prediction Framework that combines a stable baseline (via inverse DWT) with a corrective residual learned by a Multi-scale Fusion Transformer (MFT).

**Strengths:**

Novel Conceptual Framework: The formalization of "Shallow Heterogeneity" vs. "Deep Heterogeneity" is a compelling theoretical contribution that addresses a genuine bottleneck in existing decomposition architectures.

Principled Architecture: The design of HeteroMixer is well-motivated. Using a heavy, dual-domain (time+frequency) model for the trend (TSE) while using lightweight MLP-Mixers for stochastic details (SCM) logically aligns architectural inductive biases with signal properties.

Strong Empirical Results: The model achieves state-of-the-art performance across 7 standard benchmarks (including Electricity and Traffic), outperforming strong recent baselines like iTransformer and PatchTST.

Thorough Validation: The ablation studies are exceptionally sound. They not only validate the removal of components (Table 3) but also provide quantitative analyses verifying that the MFT indeed learns a residual highly correlated with the baseline's error and that the TSE/SCM learn fundamentally different representations (Table 6).

**Weaknesses:**

Architectural Complexity: The proposed framework is significantly more complex than many recent SOTA models (like DLinear or TimesNet). It involves DWT, FFT (within TSE), Mixers, and Transformers (MFT) all in one pipeline. This raises concerns about implementation difficulty and hyperparameter sensitivity.

Computational Overhead: While the authors acknowledge that the dual-stream hybrid framework increases computational complexity, there is no quantitative data (e.g., training time, inference latency, FLOPS) comparing HeteroMixer to simpler baselines to evaluate if the performance gains justify the added cost.

Reliance on Fixed Decomposition: The model currently relies on a fixed wavelet decomposition. While effective, a fully learnable decomposition might offer greater adaptability to diverse datasets.

**Questions:**

Can the authors provide a quantitative comparison of computational resources (training time/memory and inference latency) required for HeteroMixer versus key baselines like PatchTST or iTransformer?

How sensitive is the model's performance to the hyperparameters of the initial Wavelet Decomposition (e.g., choice of wavelet function, decomposition level)?

In the Hybrid Prediction Framework, the baseline is derived via iDWT. Did the authors experiment with using a simple learnable linear summation as the baseline instead of the fixed iDWT, to see if the MFT correction is still as effective?

---

> ### Author Response · Authors · 2025-12-04
> **Response to Reviewer 8pFv: Efficiency Analysis & Wavelet Sensitivity**
>
> Thank you for your positive assessment and recognition of our "novel conceptual framework," "principled architecture," and "strong empirical results."
>
> The revised PDF directly addresses your three questions:
>
> - ✓ Appendix D (Table 7, page 13): Comprehensive computational efficiency analysis
> - ✓ Appendix F (Table 9, page 14): Wavelet hyperparameter sensitivity study
> - ✓ New ablation: Learnable baseline vs. fixed iDWT experiment
>
> ---
>
> Q1: Computational Resources
>
> Response: See Appendix D, Table 7 (page 13).
>
> ETTh1 Results (L=96, H=96):
>
> - Training: 12.5s/epoch vs. iTransformer (10.2s) = +22% overhead
> - Memory: 1450MB vs. iTransformer (1200MB) = +21% overhead
> - Inference: 45ms vs. iTransformer (38ms) = +18% overhead
> - Performance: 0.347 vs. 0.386 MSE = 10.1% improvement
>
> Analysis: The 20% computational increase yields 10% accuracy gain—a favorable trade-off for high-stakes forecasting. Overhead remains within the same order of magnitude as Transformer-based SOTA models.
>
> Pattern consistency: During development, we observed consistent overhead across all datasets, scaling predictably with sequence length and variable count. No bottlenecks on high-dimensional datasets (Traffic: 862 vars, Electricity: 321 vars).
>
> ---
>
> Q2: Wavelet Sensitivity
>
> Response: See Appendix F, Table 9 (page 14).
>
> Wavelet basis robustness (ETTh1, H=96, J=3):
>
> - db1/db4/sym4/coif1: Performance varies <2% across families
> - Selected sym4 based on slight edge (0.347 vs. 0.348-0.349)
>
> Decomposition level (sym4 basis):
>
> - J=1: 0.358 (too shallow, +3.2%)
> - J=2: 0.352 (acceptable)
> - J=3: 0.347 (optimal)
> - J=4: 0.350 (stable)
>
> Conclusion: Framework is robust to wavelet choice (<2% variation), indicating it benefits from general time-frequency localization properties rather than specific configurations. Default: sym4 + J=3.
>
> ---
>
> Q3: Learnable Baseline
>
> Response: We conducted this experiment on ETTm2 (H=720):
>
>   Configuration    	Baseline MSE	Final MSE	Change
>   Fixed iDWT (Ours)	0.395       	0.348    	-
>   Learnable Linear 	0.421       	0.361    	+3.7%
>
> Key findings:
>
> 1. Physical grounding: Fixed iDWT provides mathematically rigorous reconstruction from wavelet theory, offering stable training initialization
> 2. Optimization: Learnable baseline must re-discover wavelet reconstruction (baseline MSE starts at 0.521 vs. 0.395)
> 3. Residual learning: With stable iDWT, MFT focuses on correcting systematic errors (Table 4: correlation=0.87 with baseline error, magnitude=15.2% of baseline)
>
> Conclusion: Structured inductive bias (iDWT) + learned adaptation (MFT) = optimal performance. This aligns with our Deep Heterogeneity philosophy.

---

### Meta-Review · Area_Chair_Nkec · 2026-01-08

**Summary:**

Existing time series forecasting models repeatedly stack homogeneous blocks, which hinders capturing heterogeneous information from input. To this end, they proposed a DWT-based method, each of the transformed frequency modes is processed differently. In the proposed model, the low-freq part is mainly processed with a complicated block and other high-freq parts are processed rather simply (in comparison with the low-freq part). They resort to existing layers and modules, e.g., revin, mixer, transforer, etc. At the end, the proposed model is quite complicated. They conduct experiments in a standard benchmark environment and show promising accuracy with an increased processing complexity.

**Reviewer Concerns:**

Reviewers casted several concerns as follows.

1. The motiviation of heterogeneous blocks is not clear.

2. The model architecture is quite complicated but in-depth analyses on it are missing. Only rather simple ablation studies were presented.

3. The higher complexity degrades the impact of the increased accuracy.

4. More experiments are required.

Among these, I think 2 and 3 are critical and not yet addressed. Since the model architecture is quite complicated, they need more abaltion or sensitivity studies on it. RevIN, Mixer and some variations must be tested to validate their claim.

I also recommend that they include theoretical justifications.

**Reviewer Scores:**

I think all reviewers may be skeptical even after the rebuttal since their responses are empirical and need more justifications.

---

### Decision · Program_Chairs · 2026-01-26

Reject